# OpenReview forum: "Probing Hidden Knowledge Holes in Unlearned LLMs"
_NeurIPS.cc/2025/Conference — NeurIPS 2025 poster_

### Official Review · Reviewer_A6Wn · 2025-06-30

**Clarity:** 4
**Significance:** 2
**Originality:** 3
**Rating:** 3
**Confidence:** 3

**Summary:**

This paper investigates how recent machine unlearning (MU) techniques for LLMs can inadvertently erase adjacent knowledge, unrelated to the target forget set. The authors propose a three-part probing framework: 1) identifying adjacent knowledge (non-hazardous); 2) RL-based policy to probe adjacent knowledge. and 3) method to filter the overlapping prompts. They show empirically all unlearning methods maintain performance on standard benchmarks but suffer dramatic failures on the generated probe sets. They propose and experiment with two approaches to mitigate this limitation, but they show they are not optimal solutions.

**Questions:**

*  How sensitive are the results to the choice of judge model? If a different LLM (or even a human annotator) were used to score responses, would the set of identified knowledge holes and the severity change significantly?
* You focus on factual Q&A prompts. Do you expect similar hidden forgetting in other types of tasks or domains (e.g. summarization, dialogue, code)?
* The one-shot and iterative fine-tuning strategies reintroduce old issues (harmful content or new holes). Have you considered alternative mitigation approaches (e.g. adversarial training) that might better preserve benign knowledge without reactivating the removed data?

**Ethical Concerns:**

["NO or VERY MINOR ethics concerns only"]

**Final Justification:**

The "onion effect" has been studied from a privacy perspective before (the authors do not refer to the work by Carlini, Nicholas, et al. "The privacy onion effect: Memorization is relative." Advances in Neural Information Processing Systems 35 (2022): 13263-13276.). Exact/approximate unlearning changes the input distribution; and it is interesting what biases arise under different distributions. However, this has been studied from different angles, and a comparison is lacking.
Furthermore, the new D_{AP} results after unlearning have large variance (even overlapping with the original model); weakening the empirical claim.

**Limitations:**

Some limitations are not discussed. First, the entire evaluation relies on an automated LLM scorer/reward; it is important to understand how the reward function changes the scores. Second, all probes are in the format of open-ended questions. It is unclear whether unlearning might similarly affect other capabilities (e.g. conversational or creative tasks).

**Paper Formatting Concerns:**

No formatting concerns.

**Quality:**

3

**Strengths And Weaknesses:**

Strengths:

* The paper addresses an important, under-explored problem at the intersection of LLM safety and unlearning. Introducing the notion of knowledge holes is a novel insight that highlights blind spots in current unlearning benchmarks.
* The proposed probing framework is comprehensive: it combines targeted adjacent tests (via keyword extraction) with an RL-driven discovery of latent gaps, ensuring a broad coverage of failure modes
* The methodology is well-motivated and clearly explained

Weaknesses:
* The approach depends heavily on an automated reward system (i.e. LLM as a judge). This can introduce bias; details on the calibration procedure and any biases observed should be clearly explained.
* The experiments focus on QA-style prompts; it’s unclear if similar knowledge holes would appear in other tasks (e.g. summarization or code generation)
* The mitigation strategies, although analyzed, are more illustrative than fully effective solutions – both introduce their own problems (re-adding harmful content or new holes). Furthermore, confidence bounds (standard deviation) should be computed

---

> ### Author Rebuttal · Authors · 2025-07-31
>
> We sincerely thank all reviewers for their detailed and constructive feedback. We are encouraged that they found our work to address a timely, important, and underexplored problem (b9QX, A6Wn) with an innovative and comprehensive framework (b9QX, hrSN, 1Bz9, A6Wn). We also appreciate the positive notes on our clear presentation (b9QX, hrSN, 1Bz9, A6Wn). We will provide a detailed response to address each reviewer's concerns below.
>
> ---
>
> **W1 & Q4. Dependence on LLM Judge and Sensitivity**
>
> Thank you for this question regarding the sensitivity of our results to the choice of judge model. To directly investigate this, we conducted a new set of experiments, following the other reviewers' suggestion. Specifically, we used full-parameter unlearned models, which were generated from three different seeds. We then evaluated their responses using three different judge models: gpt-4o-mini, gpt-4.1-mini, and o3-mini.
>
> The results lead to a clear conclusion. While the absolute scores show minor variations across the judges, our key findings are consistent. Specifically, across all three judge models, performance on both $D_{AP}$ and $D_{LP}$ is degraded after unlearning, compared to the original, pretrained model. Moreover, all three judges show that our latent probing set, $D_{LP}$, reveals significant knowledge holes.
>
> | Probing set | $D_{AP}$ | $D_{AP}$ | $D_{AP}$ | $D_{LP}$ | $D_{LP}$ | $D_{LP}$ | MT-bench |
> | :---: | :---: | :---: | :---: | :---: | :---: | :---: | :---: |
> | **Judge model** | gpt-4o-mini | gpt-4.1-mini | o3-mini | gpt-4o-mini | gpt-4.1-mini | o3-mini | gpt-4 |
> | **Before** | 7.33 ± 0.12 (0.00%) | 7.61 ± 0.05 (0.00%) | 7.95 ± 0.03 (0.62%) | 6.25 ± 0.11 (0.56%) | 6.51 ± 0.07 (0.33%) | 6.50 ± 0.16 (2.11%) | 6.70 |
> | **After** | 5.24 ± 2.05 (24.22%) | 5.45 ± 2.13 (22.36%) | 5.78 ± 2.11 (21.46%) | 2.36 ± 0.38 (61.30%) | 2.44 ± 0.49 (58.67%) | 2.51 ± 0.45 (58.30%) | 6.62 ± 0.47 |
>
> ---
>
> We will add the details in our revised version.
>
> **W2 & Q5. Focus on Q&A-style prompts and other tasks**
>
> Thank you for this clarifying question. Our findings do indeed extend beyond the Q&A format. As detailed in our paper, we intentionally evaluated RMU, which unlearns knowledge from a corpus of biological documents, not question-answer pairs. By showing that our probing method successfully identifies severe knowledge holes in the RMU-unlearned model, we demonstrate that this failure mode is a general side effect of unlearning, not an artifact of the Q&A format. Exploring other tasks, like code generation, is a valuable future direction for unlearning. However, we believe this is beyond the scope of our paper.
>
> ---
>
> **W3 & Q6. Illustrative Mitigation Strategies and Confidence Bounds**
>
> Thank you for your suggestion and clarifying questions. We would like to clarify that our paper's primary contribution is to diagnose the problem of knowledge holes, not to propose a complete solution. As Reviewer b9QX noted, our finding on the onion effect---where fixing one hole reintroduces harmful content or creates new holes---is itself a crucial scientific insight and motivates the need for advanced mitigation strategies.
>
> Regarding confidence bounds, our new experiments presented above were conducted over 3 independent seeds and showed the standard deviations across different runs. More importantly, we respectfully argue that the failure rate (i.e., the frequency of responses scoring 1) provided in our paper is a more crucial metric. This is because it directly quantifies the degradation in model performance compared to the original model, offering a more direct measure of knowledge loss.
>
> We hope these clarifications and additional experiments address your concerns. We would be happy to discuss any further questions or comments.

---

> > ### Comment · Reviewer_A6Wn · 2025-08-08
> >
> > Thank you for the rebuttal. We are in agreement that the study of the "onion effect" is important. This has been studied from a privacy perspective before (please see Carlini, Nicholas, et al. "The privacy onion effect: Memorization is relative." Advances in Neural Information Processing Systems 35 (2022): 13263-13276.). Exact/approximate unlearning changes the input distribution; and it is interesting what biases arise under different distributions. However, this has been studied from different angles, and a comparison with existing literature will enrich your work. Furthermore, the D_{AP} scores after unlearning have large variance (even overlapping with the original model); weakening the empirical claim. In light of these, I'd like to maintain my score.

---

> > > ### Author Response · Authors · 2025-08-09
> > > **Official Comment by Authors**
> > >
> > > Thank you for highlighting the connection to Carlini et al.'s work on the privacy onion effect. While we acknowledge this prior work, our study ****differs substantially**** in several key dimensions. Carlini et al. focused on image classification tasks (CIFAR-10/100) and investigated how direct sample removal affects privacy risk as measured by membership inference attacks.
> > >
> > > In contrast, ****our work examines large language models and studies how indirect knowledge removal through an unlearning algorithm affects the preservation of benign knowledge—a fundamentally different type of side effect****. We borrowed the "onion" terminology to illustrate the iterative nature of how addressing knowledge holes can create new ones, but the underlying phenomenon we study (unintended loss of benign knowledge during targeted harmful content removal) is distinct from the privacy memorization effects examined by Carlini et al.
> > >
> > > We also want to emphasize that ****the "onion effect" represents just one finding among several contributions in our paper****. Our primary contribution is the first principled framework for dynamically discovering knowledge holes in unlearned LLMs, which addresses a previously unaddressed evaluation gap in the unlearning literature.
> > >
> > > Regarding the $D_\text{AP}$ variance, we acknowledge the overlapping confidence intervals while noting that ****our statistical significance tests consistently demonstrate meaningful degradation across judge models****. Specifically, we conducted independent two-sample t-tests to verify the statistical significance of the performance drop. Across all three judge models, the difference was found to be statistically significant (gpt-4o-mini: p=0.016; gpt-4.1-mini: p=0.016; o3-mini: p=0.015) even for full-parameter unlearning.
> > >
> > > More importantly, ****our RL-based latent probing ($D_\text{LP}$) reveals severe knowledge holes regardless of $D_\text{AP}$ variance****, which we believe validates the necessity of dynamic evaluation frameworks for real-world unlearning deployments.

---

> ### Author Response · Authors · 2025-08-06
> **Further Clarifications on W2, Q5 and Limitations.**
>
> We would like to further clarify our focus on the Q&A format. This format is a direct result of our framework's design: we use our $D_{adj}$ set to seed an RL policy, which then continuously generates Q&A-style latent probes to find knowledge holes during training.
>
> In response to the reviewer's point, we agree that knowledge holes could occur in other domains. However, reliably probing knowledge holes in specialized tasks such as code generation or summarization is challenging. For example, accurately measuring knowledge degradation in code generation requires a proficient victim model; a model that performs poorly on this task could yield misleading results after unlearning. Our selection of victim models was guided by two practical considerations. First, the models needed a light moderation level to ensure they could generate the harmful responses we aimed to unlearn. Second, for the bio-corpus unlearning task (e.g., WMDP), we used their unlearned models to ensure a direct and valid evaluation. Furthermore, probing summarization would require a policy model capable of overcoming the difficulty of generating diverse and long-context test cases. Therefore, we chose to diagnose the degradation of general, user-facing knowledge with a Q&A format probing set. More importantly, we found this approach to be both direct and effective, as it consistently revealed significant knowledge holes for general and benign queries in unlearned models.
>
> Probing hidden degradation in a specific task, perhaps by designing a reward signal with an appropriate policy model and seed initialization, is an interesting direction for future work. We thank the reviewer again for raising this important point. We will add this, along with a discussion on our reliance on the LLM-as-a-judge, to the Limitations section of our revised paper.
>
> We hope this further addresses the reviewer's concern. Should the reviewer have any remaining concerns, we would be happy to discuss them.

---

> ### Author Response · Authors · 2025-08-07
> **Official Comment by Authors**
>
> Dear Reviewer A6Wn,
>
> As the discussion period will be closing soon, we would like to kindly follow up and ask if you have any remaining concerns.
>
> We believe we have addressed the concerns you raised in our rebuttal. For instance, to address your point regarding the sensitivity of the LLM judge, we ran a new set of experiments with three different judge models. We found that our key findings are consistent across all judge models. Moreover, we provided the standard deviations from the new experiments, and further clarified the importance of our failure rate metric for measuring knowledge loss.
>
> In addition to these experimental updates, we explained our rationale for focusing on the Q&A format and our perspective on mitigation.
>
> We are grateful for your time and feedback, which have helped us strengthen the paper. Please let us know if you have any other questions.
>
> Best regards,
>
> The Authors of Submission 25003

---

### Official Review · Reviewer_1Bz9 · 2025-07-02

**Clarity:** 3
**Significance:** 4
**Originality:** 3
**Rating:** 4
**Confidence:** 4

**Summary:**

This paper studies the effect of LLM unlearning on the models’ performance on some “benign” datasets that are not captured by current unlearning benchmarks. The paper introduces two ways to generate adversarial datasets for which the LLM performance downgrades. The first method finds neighbours (D_adj) of the forget-set that are “benign” yet the model’s performance is low. Second method uses RL to learn a policy to generate broader examples (D_RL) for which the model’s performance is low. These sets are gone through some post-hoc filtering to remove invalid examples. The results show that the original (pre-unlearning) model’s performance on these datasets is indeed higher than the unlearned model’s performance.

**Questions:**

please see the weaknesses.

**Ethical Concerns:**

["NO or VERY MINOR ethics concerns only"]

**Final Justification:**

The paper studies an interesting problem, i.e. understanding the impacts of unlearning on the downstream model. I believe this method can help creating more comprehensive unlearning benchmarks.

Two of my concerns, namely, comparisons with other adversarial methods and robustness of LLM judge remains open and not fully resolved during the rebuttal. But I still believe the findings of the paper have potential future impact.

**Limitations:**

yes

**Paper Formatting Concerns:**

Appendix E is referenced but does not exist. it should be D.

**Quality:**

3

**Strengths And Weaknesses:**

**Strength**

**S1**: Showing where/when unlearning fails is very important. Recent similar works usually look at the failures of unlearning w.r.t the forget-set and the potential to recover that knowledge. The hidden effect on utility is not addressed as much

**S2**: The ideas are simple to follow and are well presented.

**S3**: This method can help creating more comprehensive unlearning benchmarks.

**Weaknesses**


**W1**:  All the unlearning baselines are performed on low-rank parameters instead of the full model’s parameters. I wonder if the results repeat in that case?

**W2**: Something is not clear to me. Do you generate D_RL and then used it to evaluate both original model and the unlearned models? Under a fairer comparison a similar D_RL should be generated for the original model, separately.

**W3**: From a higher level D_adj and D_RL are created using some adversarial methods. There are other adversarial methods that finds prompts that make the model hallucinate.  There is no comparisons with these methods. Do the author think it’s impossible to compare against them? Why?

**W4**: The VENDI score reported in Table 4 are usually below 50%. What a good VENDI score should look like? Any reference point?

**W5**: I suggest scaling the results of Table 1 by the diversity scores in Table 2. This way we get a purer picture of the performance of the model on this “knowledge holes”

**W6**: the paper uses LLMs for judging and generating frequently. However, no evidence about the robustness of these LLMs for these usecases is reported. For example in the case of LLM judge you could evaluate the accuracy on let’s say 100 of examples, manually. Or repeat the experiments with a different LLM and report the differences.

**W7**: The authors admit that error bars are not reported. While I understand the cost of this, but I’d appreciate some variance number at least for a couple of experiments to get some insights about the noise.

**W8**: D_AP and D_LP are not formally introduced.

**W9**: Appendix E doesn’t exist. It should be Appendix D, right?

---

> ### Author Rebuttal · Authors · 2025-07-31
>
> We sincerely thank all reviewers for their detailed and constructive feedback. We are encouraged that they found our work to address a timely, important, and underexplored problem (b9QX, A6Wn) with an innovative and comprehensive framework (b9QX, hrSN, 1Bz9, A6Wn). We also appreciate the positive notes on our clear presentation (b9QX, hrSN, 1Bz9, A6Wn). We will provide a detailed response to address each reviewer's concerns below.
>
> ---
>
> **W1. Unlearning on low-rank (LoRA) vs. full-model parameters**
>
> Thank you for raising this important question. To address concerns about LoRA, we have run new experiments for the LLMU method using full-parameter finetuning. To further substantiate the results, we performed unlearning with three different seeds (with a smaller learning rate and iterations due to the sensitivity of the unlearning on full parameters), generated responses for $D_{AP}$ and $D_{LP}$ over three independent runs, and evaluated all responses with three different judge models. The results, shown below, demonstrate that the knowledge hole phenomenon is not an artifact of LoRA and is indeed generalizable. Specifically, performance on $D_{AP}$ and $D_{LP}$ is consistently degraded compared to the original model. Moreover, $D_{LP}$ consistently widens the performance gap compared to $D_{AP}$, confirming the necessity of our dynamic probing method. The results below are presented as mean ± standard deviation across the different runs.
>
> | Probing set | $D_{AP}$ | $D_{AP}$ | $D_{AP}$ | $D_{LP}$ | $D_{LP}$ | $D_{LP}$ | MT-bench |
> | :---: | :---: | :---: | :---: | :---: | :---: | :---: | :---: |
> | **Judge model** | gpt-4o-mini | gpt-4.1-mini | o3-mini | gpt-4o-mini | gpt-4.1-mini | o3-mini | gpt-4 |
> | **Before** | 7.33 ± 0.12 (0.00%) | 7.61 ± 0.05 (0.00%) | 7.95 ± 0.03 (0.62%) | 6.25 ± 0.11 (0.56%) | 6.51 ± 0.07 (0.33%) | 6.50 ± 0.16 (2.11%) | 6.70 |
> | **After** | 5.24 ± 2.05 (24.22%) | 5.45 ± 2.13 (22.36%) | 5.78 ± 2.11 (21.46%) | 2.36 ± 0.38 (61.30%) | 2.44 ± 0.49 (58.67%) | 2.51 ± 0.45 (58.30%) | 6.62 ± 0.47 |
>
> ---
>
> **W2. Fairer comparison of $D_{LP}$**
>
> Thank you for your clarifying question. Our goal is to identify and quantify the 'knowledge holes' that are specifically created by the unlearning process, thereby measuring what knowledge the unlearned model has lost compared to its original state.
>
> To achieve this, the test set ($D_{LP}$) is generated using the failure signals from the unlearned model. We then evaluate these discovered prompts on the original model for the crucial reason of confirming that these questions were indeed answerable before unlearning.
>
> The resulting performance gap between the two models clearly demonstrates the knowledge compromised by the unlearning procedure. We note that our goal is not to diagnose pre-existing weaknesses in the original model, but to isolate the specific side effects of the unlearning process itself.
>
> ---
>
> **W3. Comparison with other adversarial methods (e.g., finding hallucinations)**
>
> Thank you for this question. We would like to clarify that our goal is not to conduct a comparative study of which adversarial method finds the most failures. Rather, our contribution is to propose a systematic probing framework designed to evaluate a hidden risk from unlearning that prior works have overlooked. This diagnostic goal is distinct from hallucination-finding, which targets plausible but factually incorrect statements.
>
> ---
>
> **W4. Reference point for VENDI score**
>
> Thank you for the question. Generally, a higher VENDI score is better. To provide a reference point, Table 2 of our paper shows the VENDI scores for standard benchmarks like MT-bench (0.519) and MMLU (0.274). Our generated $D_{LP}$ sets achieve scores (0.35-0.43) that are comparable or higher, demonstrating significant semantic diversity. Furthermore, we provide the diversity score to demonstrate that our framework discovers a sufficiently diverse set of failures and does not simply converge on a few simple failure modes.
>
> ---
>
> **W5. Scaling results by diversity scores**
>
> Thank you for your suggestion. We present the scores separately to offer a more intuitive and complete analysis, as our goal is to highlight two distinct aspects of the knowledge hole problem. First, the judge score (Table 1) measures the severity of the failure on an individual prompt---i.e., how poor the model's response is. In parallel, the diversity score (Table 2), as clarified in our response to W4, demonstrates that our probing discovers a wide range of failures and does not simply converge on a few simple failure modes. Combining these two different dimensions into a single scaled score could obscure this crucial understanding of our work.
>
> ---
>
> **W6 & W7. Robustness of LLM Judge and Lack of Error Bars**
>
> Thank you for these important points. We address both with our new experimental results presented in the table for W1. Specifically: 1) Judge Robustness: By evaluating with three distinct judge models and finding highly consistent trends, we demonstrate our findings are not sensitive to a single judge's bias. 2) Error Bars: By running the new experiments with three different seeds, we now report the mean and standard deviation, providing the requested insight into the variance of our findings. More specifically, these results show that our RL-based dynamic probing is stable and consistently effective at uncovering severe knowledge holes.
>
> ---
>
> **W8. Formal introduction of $D_{AP}$ and $D_{LP}$**
>
> We appreciate the question regarding formal definitions. Our approach intentionally uses a procedural rather than formal definition because our focus is on the practical generation of test cases. In particular, adjacent knowledge consists of benign prompts that share certain keywords or concepts with the forgetting set. Latent Knowledge refers to benign prompts that do not share these keywords, discovered via our RL-based probing. Our primary contribution is an empirical framework for uncovering knowledge holes. For that goal, a procedural definition clearly articulates how each prompt set is constructed and why it targets different types of unintended model failures. Also, this definition ensures that other researchers can replicate our procedure and arrive at the same test sets. We believe this approach offers sufficient rigor and clarity for evaluating unlearning methods, though future work could further formalize these concepts if desired.
>
> ---
>
> **W9. Appendix E typo**
>
> Thank you for your careful reading. We will correct the reference to Appendix D in our revised paper.
>
> We hope these clarifications and additional experiments address your concerns. We would be happy to discuss any further questions or comments.

---

> > ### Comment · Reviewer_1Bz9 · 2025-08-02
> >
> > Thanks for the clarifications and the extra results. For now I maintain my score.

---

### Official Review · Reviewer_hrSN · 2025-07-03

**Clarity:** 3
**Significance:** 1
**Originality:** 2
**Rating:** 4
**Confidence:** 5

**Summary:**

This paper proposes a reinforcement learning approach to generate natural language prompts for unlearned models that surface poor performance on information that should be 'retained' after unlearning. The datasets generated using this approach successfully demonstrate that popular unlearning methods fail to preserve coherent answers on benign queries that are related to the forget data.

**Questions:**

The paper was clear. I do not have any specific clarifications that would be addressable during the rebuttal phase or would change my score.

**Ethical Concerns:**

["NO or VERY MINOR ethics concerns only"]

**Final Justification:**

The concerns from my review still stand regarding whether the paper adds significant understanding over what is already known about the brittleness of unlearning algorithms. However I have maintained my score due to the potential utility of the benchmark.

**Limitations:**

Yes

**Quality:**

2

**Strengths And Weaknesses:**

Strengths:
- The authors propose a new way to generate unlearning evaluation corpora that more thoroughly covers the benign prompt space compared to prior work in unlearning evaluation.
- The results are fairly dramatic in that models perform poorly on a majority of prompts in the generated prompt set.

Weaknesses:
- My main overall concern is that at this point it is fairly well known that unlearning algorithms, especially the ones studied in this work, are brittle and perform poorly under small perturbations to the retain (or forget) corpus. As such, although this work more thoroughly explores that space of perturbations, it is not clear what new insight a reader gains about the field of unlearning by reading this paper.
- Moreover, algorithms cannot yet perform well even under smaller perturbations than the ones proposed in this paper.
- Automated generation of natural prompts to elicit undesirable behavior has been studied by, for example, https://arxiv.org/pdf/2502.01236 . Brittleness of unlearning methods under small perturbations to benchmarks has been studied by, for example, https://arxiv.org/pdf/2410.02879 as well as RWKU that is cited by the authors. While these are not as exhaustive as what is proposed in this work, my main concern is that the work here does not add much to our understanding of unlearning as compared to these prior works.

---

> ### Author Rebuttal · Authors · 2025-07-31
>
> We sincerely thank all reviewers for their detailed and constructive feedback. We are encouraged that they found our work to address a timely, important, and underexplored problem (b9QX, A6Wn) with an innovative and comprehensive framework (b9QX, hrSN, 1Bz9, A6Wn). We also appreciate the positive notes on our clear presentation (b9QX, hrSN, 1Bz9, A6Wn). We will provide a detailed response to address each reviewer's concerns below.
>
> ---
>
> **W1, W3: Contribution and Novelty (Distinction from Prior Work)**
>
> We thank the reviewer for thoughtful feedback and for their broad understanding of prior work in machine unlearning. We respectfully argue that our work's primary contribution is not to simply re-demonstrate the brittleness of the current unlearning methods, but to introduce a probing framework for a different and more insidious class of failure that prior work does not systematically address.
>
> Specifically, prior works [1, 2] analyze perturbations that are linked to the forget set (e.g., neighbor knowledge or combining forget/retain prompts). In contrast, our work identifies a different failure mode: damage to model utility triggered by benign prompts that are semantically unrelated to the forgotten content and are generated model-dependently. This approach reveals hidden risks that methods focused on predefined or neighboring test cases cannot fully capture.
>
> Moreover, the work in [3] is a general red-teaming tool for eliciting undesirable behaviors. Our goal is different and specifically tailored to probing the unlearned models. We diagnose the loss of benign capability by finding prompts that elicit low-quality or nonsensical responses to benign questions.
>
> Therefore, our work provides an important diagnostic tool that complements static benchmarks by adaptively discovering how and where each unlearning process uniquely damages a model's general knowledge. This quantifies a non-trivial and previously overlooked risk, offering a critical insight for developing more robust unlearning methods and contributing a new dimension for diagnosis to the unlearning community.
>
> ---
>
> **W2: Relevance of Probing Given Failures on Simpler Perturbations**
>
> Thank you for your comment. We agree that unlearning can fail on smaller perturbations. However, our method is not simply a harder test for this known instability. Rather, it provides a different angle on the risks of unlearning by probing a broader, more insidious class of failures. While prior tests primarily reveal the local instability of the unlearning process, our work offers a more comprehensive diagnostic perspective, capable of assessing both local side effects (through our adjacent knowledge probing) and the broader, latent nature of unintended knowledge loss. A deeper understanding of these diverse failure angles will help future research to more thoroughly design and evaluate unlearning methods.
>
> We hope this addresses your concerns. We would be happy to discuss any further questions or comments.
>
> ---
>
> ### References
> 1. Position: LLM Unlearning Benchmarks are Weak Measures of Progress
> 2. RWKU: Benchmarking Real-World Knowledge Unlearning for Large Language Models
> 3. Eliciting Language Model Behaviors with Investigator Agents

---

### Official Review · Reviewer_b9QX · 2025-07-09

**Clarity:** 3
**Significance:** 4
**Originality:** 3
**Rating:** 5
**Confidence:** 4

**Summary:**

This paper investigates the extent to which unlearning methods introduce unwanted "knowledge holes" in LLMs, outside the domain targeted by unlearning. They propose an RL-based framework for generating test cases to probe for these knowledge holes, and use an LLM judge to determine whether the unlearned model passes or fails a particular test case (that is, whether it has lost the knowledge needed to appropriately respond to the prompt). They find that all four tested unlearning methods introduce substantial knowledge holes which are obscured by naive untargeted benchmarks.

**Questions:**

1. Is there prior work exploring how the usage of LoRA affects the efficacy of unlearning? Presumably LoRA increases retention of pretraining knowledge relative to full finetuning, but the effects could be complex and nonlinear.
2. How often do you think the knowledge holes you find would be encountered in practice?

**Ethical Concerns:**

["NO or VERY MINOR ethics concerns only"]

**Final Justification:**

The additional experiments with full fine tuning address my primary concern with the initial submission. It also seems that the kinds of knowledge holes discovered by this approach are fairly likely to be encountered in natural interactions with the language model and are therefore worth detecting.

**Limitations:**

yes

**Quality:**

3

**Strengths And Weaknesses:**

Strengths:
- Timely and neglected topic, clear presentation
- Use of RL to search for test cases is innovative and seems to uncover more failing cases
- Demonstrates that mitigation of knowledge holes is a difficult whack-a-mole game

Weaknesses:
- LoRA is used for all experiments, which may limit the generalizability of the results to full finetuning. Also, this detail is not mentioned in the body. I am uncertain how much this affects the results, but it is plausible that the effect could be substantial. Replicating the results without LoRA would cause me to raise my score.
- The base models used should probably be listed in the body, rather than the Appendix, since this is an important hyperparameter as well.

---

> ### Author Rebuttal · Authors · 2025-07-31
>
> We sincerely thank all reviewers for their detailed and constructive feedback. We are encouraged that they found our work to address a timely, important, and underexplored problem (b9QX, A6Wn) with an innovative and comprehensive framework (b9QX, hrSN, 1Bz9, A6Wn). We also appreciate the positive notes on our clear presentation (b9QX, hrSN, 1Bz9, A6Wn). We will provide a detailed response to address each reviewer's concerns below.
>
> ---
>
> **W1, Q1. Generalizability (LoRA vs. Full-parameter finetuning)**
>
> Thank you for raising this important question regarding the generalizability of our findings beyond LoRA. In our preliminary tests, we found that full-parameter unlearning is highly sensitive; for instance, using the same learning rate (i.e., 5e-5) as in our LoRA experiments, the unlearning loss often reached its maximum threshold within just a few iterations. Therefore, we ran new experiments for the LLMU method using full-parameter finetuning but with a smaller learning rate (i.e., 1e-6).
>
> To further substantiate the results, we performed unlearning with three different seeds. From each of these three models, we then conducted three independent response generation runs for $D_{AP}$ and $D_{LP}$, and evaluated all responses with three different judge models. The results, shown below, demonstrate that the knowledge hole phenomenon is not an artifact of LoRA and is indeed generalizable. Specifically, performance on $D_{AP}$ and $D_{LP}$ is consistently degraded compared to the original model. Moreover, $D_{LP}$ consistently widens the performance gap compared to $D_{AP}$, confirming the necessity of our dynamic probing method. The results below are presented as mean ± standard deviation across the different runs. Percentages indicate the proportion of prompts that lead to scores of 1 on the response.
>
> | Probing set | $D_{AP}$ | $D_{AP}$ | $D_{AP}$ | $D_{LP}$ | $D_{LP}$ | $D_{LP}$ | MT-bench |
> | :---: | :---: | :---: | :---: | :---: | :---: | :---: | :---: |
> | **Judge model** | gpt-4o-mini | gpt-4.1-mini | o3-mini | gpt-4o-mini | gpt-4.1-mini | o3-mini | gpt-4 |
> | **Before** | 7.33 ± 0.12 (0.00%) | 7.61 ± 0.05 (0.00%) | 7.95 ± 0.03 (0.62%) | 6.25 ± 0.11 (0.56%) | 6.51 ± 0.07 (0.33%) | 6.50 ± 0.16 (2.11%) | 6.70 |
> | **After** | 5.24 ± 2.05 (24.22%) | 5.45 ± 2.13 (22.36%) | 5.78 ± 2.11 (21.46%) | 2.36 ± 0.38 (61.30%) | 2.44 ± 0.49 (58.67%) | 2.51 ± 0.45 (58.30%) | 6.62 ± 0.47 |
>
> ---
>
> **W2. Base models listed in the body**
>
> We appreciate the reviewer’s attention to detail. We agree that this is an important experimental detail and will move the base model information from the Appendix to the main body of the paper for better clarity.
>
> ---
>
> **Q2. Practical frequency of knowledge holes**
>
> Thank you for this insightful question. While it's challenging to quantify the exact probability of encountering a specific knowledge hole, our key finding is that these failures are triggered by benign and common questions that a user would expect any capable assistant to handle.
>
> To illustrate this, we provide five diverse examples below, drawn from our new full-parameter unlearning experiments. For a more systematic analysis of the broad topics these failures cover, we refer to our clustering analysis in Appendix C.5 (Tables 13-16).
>
> #### Qualitative Examples of Latent Knowledge Holes.
>
> | Category                        | Example Question                                                    |
> | :------------------------------ | :------------------------------------------------------------------ |
> | **Health & Safety** | How do you know when you're having a stroke?                      |
> | **Daily Life & Practical Advice** | What's the best way to stop smoking?                              |
> | **Household Chores** | What is the best way to clean a toilet?                           |
> | **Geography & World Knowledge** | Which two countries share a border but have no land connection?   |
> | **Space Exploration** | How do astronauts go to the bathroom in space?                    |
>
> As these examples show, knowledge holes are not limited to rare edge cases but can cause failures on everyday queries across many domains. The practical significance of our work lies in providing a methodology to uncover this unpredictable and widespread degradation of model utility on benign, real-world questions.
>
> We hope these clarifications and additional experiments address your concerns. We would be happy to discuss any further questions or comments.

---

> ### Author Response · Authors · 2025-08-07
> **Official Comment by Authors**
>
> Dear Reviewer b9QX,
>
> As the discussion period will be closing soon, we would like to kindly follow up and ask if you have any remaining concerns.
>
> We believe we have addressed the concerns you raised in our rebuttal and new experiments. Specifically, to address your main concern about LoRA, we have conducted new experiments with full-parameter finetuning. The results demonstrate that the knowledge hole phenomenon is generalizable and not specific to the usage of LoRA.
>
> Additionally, to answer your question about the practical frequency of these issues, we have provided qualitative examples of benign, common questions that fail after unlearning, along with the clustering results in our appendix.
>
> We are grateful for your time and feedback, which have helped us strengthen the paper. Please let us know if you have any other questions.
>
> Best regards,
>
> The Authors of Submission 25003

---

### Note · Authors · 2025-08-15

We thank the reviewers for their constructive feedback, which helped us strengthen the paper. We have addressed their concerns with further experiments and clarifications. The main updates are as follows:

In response to concerns about LoRA, **we conducted new experiments using full-parameter fine-tuning**. The results confirm that performance degradation on our probing test cases is a consistent outcome, indicating that the knowledge hole is a general challenge, not an artifact of a LoRA-based approach.

To address questions of LLM sensitivity and error bars, we present results from full-parameter tuning **across multiple seeds and response generations, evaluated by three distinct LLM judges**. This comprehensive assessment demonstrates the consistency of our observations. Crucially, to resolve the concern about performance overlap, **we performed independent two-sample t-tests on $D_\text{AP}$**, verifying that the observed performance degradation is statistically significant (i.e., p <= 0.016).

We have also clarified our scope. First, **our work is not intended as a comparative benchmark against existing tools** that are not designed for our specific problem of probing benign knowledge loss from unlearning. Instead, we developed a tailored RL framework to identify the novel failure mode. Second, **the Q&A format was a deliberate choice, proving highly effective at revealing significant knowledge holes** and forming an integral part of our RL framework. Third, we clarified that exploring exhaustive mitigation strategies is beyond the current scope. Finally, for clarity, we detailed our procedural definitions, evaluation metrics, and setup.

Our work tackles a critical gap in evaluating unlearned models. Existing methods rely on static benchmarks that measure performance only on known information, leaving them blind to the unpredictable knowledge holes that emerge dynamically during unlearning. To solve this, our work introduces **a dynamic probing framework that systematically surfaces these hidden failures**, acting as a crucial complement to existing benchmarks. Moreover, we identify **an onion effect in unlearning that is fundamentally distinct from prior work** by focusing on how unlearning affects the preservation of benign knowledge.

We believe our work is an important stepping stone in bridging a critical gap in the unlearning literature, and we are grateful for the constructive process that has made the revised paper a stronger contribution.

---

### Decision · Program_Chairs · 2025-09-17

**Decision:**

Accept (poster)

**Comment:**

The reviewers agree that the paper shows new pitfalls for existing unlearning algorithms. This is useful for evaluating the existing and future unlearning methods.